# Neural Ordinary Differential Equations

**Ricky T. Q. Chen\*, Yulia Rubanova\*, Jesse Bettencourt\*, David Duvenaud**
University of Toronto, Vector Institute
{rtqichen, rubanova, jessebett, duvenaud}@cs.toronto.edu

## Abstract

We introduce a new family of deep neural network models. Instead of specifying a discrete sequence of hidden layers, we parameterize the derivative of the hidden state using a neural network. The output of the network is computed using a black-box differential equation solver. These continuous-depth models have constant memory cost, adapt their evaluation strategy to each input, and can explicitly trade numerical precision for speed. We demonstrate these properties in continuous-depth residual networks and continuous-time latent variable models. We also construct continuous normalizing flows, a generative model that can train by maximum likelihood, without partitioning or ordering the data dimensions. For training, we show how to scalably backpropagate through any ODE solver, without access to its internal operations. This allows end-to-end training of ODEs within larger models.

## 1 Introduction

Models such as residual networks, recurrent neural network decoders, and normalizing flows build complicated transformations by composing a sequence of transformations to a hidden state:

$$\mathbf{h}_{t+1} = \mathbf{h}_t + f(\mathbf{h}_t, \theta_t) \tag{1}$$

where $t \in \{0 \ldots T\}$ and $\mathbf{h}_t \in \mathbb{R}^D$. These iterative updates can be seen as an Euler discretization of a continuous transformation (Lu et al., 2017; Haber and Ruthotto, 2017; Ruthotto and Haber, 2018).

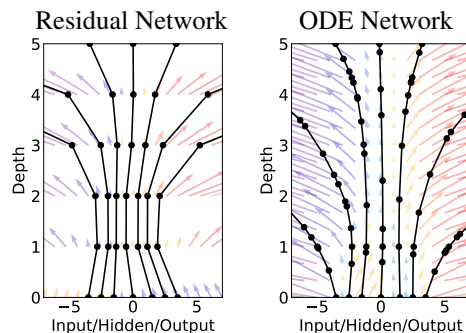

Figure 1: *Left:* A Residual network defines a discrete sequence of finite transformations. *Right:* A ODE network defines a vector field, which continuously transforms the state. *Both:* Circles represent evaluation locations.

What happens as we add more layers and take smaller steps? In the limit, we parameterize the continuous dynamics of hidden units using an ordinary differential equation (ODE) specified by a neural network:

$$\frac{d\mathbf{h}(t)}{dt} = f(\mathbf{h}(t), t, \theta) \tag{2}$$

Starting from the input layer $\mathbf{h}(0)$, we can define the output layer $\mathbf{h}(T)$ to be the solution to this ODE initial value problem at some time $T$. This value can be computed by a black-box differential equation solver, which evaluates the hidden unit dynamics $f$ wherever necessary to determine the solution with the desired accuracy. Figure 1 contrasts these two approaches.

Defining and evaluating models using ODE solvers has several benefits:

**Memory efficiency** In Section 2, we show how to compute gradients of a scalar-valued loss with respect to all inputs of any ODE solver, *without backpropagating through the operations of the solver*. Not storing any intermediate quantities of the forward pass allows us to train our models with constant memory cost as a function of depth, a major bottleneck of training deep models.

**Adaptive computation** Euler's method is perhaps the simplest method for solving ODEs. There have since been more than 120 years of development of efficient and accurate ODE solvers (Runge, 1895; Kutta, 1901; Hairer et al., 1987). Modern ODE solvers provide guarantees about the growth of approximation error, monitor the level of error, and adapt their evaluation strategy on the fly to achieve the requested level of accuracy. This allows the cost of evaluating a model to scale with problem complexity. After training, accuracy can be reduced for real-time or low-power applications.

**Parameter efficiency** When the hidden unit dynamics are parameterized as a continuous function of time, the parameters of nearby "layers" are automatically tied together. In Section 3, we show that this reduces the number of parameters required on a supervised learning task.

**Scalable and invertible normalizing flows** An unexpected side-benefit of continuous transformations is that the change of variables formula becomes easier to compute. In Section 4, we derive this result and use it to construct a new class of invertible density models that avoids the single-unit bottleneck of normalizing flows, and can be trained directly by maximum likelihood.

**Continuous time-series models** Unlike recurrent neural networks, which require discretizing observation and emission intervals, continuously-defined dynamics can naturally incorporate data which arrives at arbitrary times. In Section 5, we construct and demonstrate such a model.

## 2   Reverse-mode automatic differentiation of ODE solutions

The main technical difficulty in training continuous-depth networks is performing reverse-mode differentiation (also known as backpropagation) through the ODE solver. Differentiating through the operations of the forward pass is straightforward, but incurs a high memory cost and introduces additional numerical error.

We treat the ODE solver as a black box, and compute gradients using the *adjoint sensitivity method* (Pontryagin et al., 1962). This approach computes gradients by solving a second, augmented ODE backwards in time, and is applicable to all ODE solvers. This approach scales linearly with problem size, has low memory cost, and explicitly controls numerical error.

Consider optimizing a scalar-valued loss function $L()$, whose input is the result of an ODE solver:

$$L(\mathbf{z}(t_1)) = L\left(\mathbf{z}(t_0) + \int_{t_0}^{t_1} f(\mathbf{z}(t), t, \theta)dt\right) = L(\text{ODESolve}(\mathbf{z}(t_0), f, t_0, t_1, \theta)) \quad (3)$$

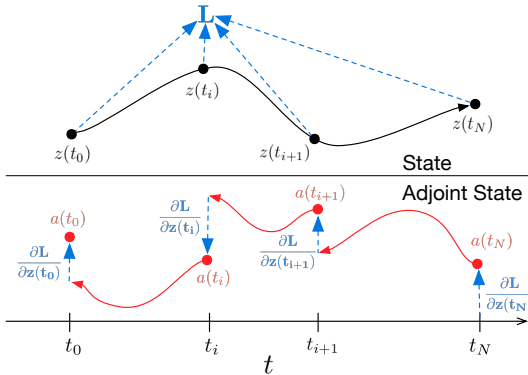

To optimize $L$, we require gradients with respect to $\theta$. The first step is to determining how the gradient of the loss depends on the hidden state $\mathbf{z}(t)$ at each instant. This quantity is called the *adjoint* $\mathbf{a}(t) = \partial L/\partial \mathbf{z}(t)$. Its dynamics are given by another ODE, which can be thought of as the instantaneous analog of the chain rule:

$$\frac{d\mathbf{a}(t)}{dt} = -\mathbf{a}(t)^{\mathsf{T}}\frac{\partial f(\mathbf{z}(t), t, \theta)}{\partial \mathbf{z}} \quad (4)$$

We can compute $\partial L/\partial \mathbf{z}(t_0)$ by another call to an ODE solver. This solver must run backwards, starting from the initial value of $\partial L/\partial \mathbf{z}(t_1)$. One complication is that solving this ODE requires the knowing value of $\mathbf{z}(t)$ along its entire trajectory. However, we can simply recompute $\mathbf{z}(t)$ backwards in time together with the adjoint, starting from its final value $\mathbf{z}(t_1)$.

Computing the gradients with respect to the parameters $\theta$ requires evaluating a third integral, which depends on both $\mathbf{z}(t)$ and $\mathbf{a}(t)$:

$$\frac{dL}{d\theta} = \int_{t_1}^{t_0} \mathbf{a}(t)^{\mathsf{T}}\frac{\partial f(\mathbf{z}(t), t, \theta)}{\partial \theta}dt \quad (5)$$

Figure 2: Reverse-mode differentiation of an ODE solution. The adjoint sensitivity method solves an augmented ODE backwards in time. The augmented system contains both the original state and the sensitivity of the loss with respect to the state. If the loss depends directly on the state at multiple observation times, the adjoint state must be updated in the direction of the partial derivative of the loss with respect to each observation.

The vector-Jacobian products $\mathbf{a}(t)^T \frac{\partial f}{\partial \mathbf{z}}$ and $\mathbf{a}(t)^T \frac{\partial f}{\partial \theta}$ in (4) and (5) can be efficiently evaluated by automatic differentiation, at a time cost similar to that of evaluating $f$. All integrals for solving $\mathbf{z}$, $\mathbf{a}$ and $\frac{\partial L}{\partial \theta}$ can be computed in a single call to an ODE solver, which concatenates the original state, the adjoint, and the other partial derivatives into a single vector. Algorithm 1 shows how to construct the necessary dynamics, and call an ODE solver to compute all gradients at once.

---

**Algorithm 1** Reverse-mode derivative of an ODE initial value problem

---

**Input:** dynamics parameters $\theta$, start time $t_0$, stop time $t_1$, final state $\mathbf{z}(t_1)$, loss gradient $\partial L/\partial \mathbf{z}(t_1)$

$s_0 = [\mathbf{z}(t_1), \frac{\partial L}{\partial \mathbf{z}(t_1)}, \mathbf{0}_{|\theta|}]$            $\triangleright$ Define initial augmented state

  **def** aug_dynamics($[\mathbf{z}(t), \mathbf{a}(t), \cdot], t, \theta$):        $\triangleright$ Define dynamics on augmented state

    **return** $[f(\mathbf{z}(t), t, \theta), -\mathbf{a}(t)^\mathsf{T} \frac{\partial f}{\partial \mathbf{z}}, -\mathbf{a}(t)^\mathsf{T} \frac{\partial f}{\partial \theta}]$     $\triangleright$ Compute vector-Jacobian products

$[\mathbf{z}(t_0), \frac{\partial L}{\partial \mathbf{z}(t_0)}, \frac{\partial L}{\partial \theta}] = \text{ODESolve}(s_0, \text{aug\_dynamics}, t_1, t_0, \theta)$     $\triangleright$ Solve reverse-time ODE

**return** $\frac{\partial L}{\partial \mathbf{z}(t_0)}, \frac{\partial L}{\partial \theta}$              $\triangleright$ Return gradients

---

Most ODE solvers have the option to output the state $\mathbf{z}(t)$ at multiple times. When the loss depends on these intermediate states, the reverse-mode derivative must be broken into a sequence of separate solves, one between each consecutive pair of output times (Figure 2). At each observation, the adjoint must be adjusted in the direction of the corresponding partial derivative $\partial L/\partial \mathbf{z}(t_i)$.

The results above extend those of Stapor et al. (2018, section 2.4.2). An extended version of Algorithm 1 including derivatives w.r.t. $t_0$ and $t_1$ can be found in Appendix C. Detailed derivations are provided in Appendix B. Appendix D provides Python code which computes all derivatives for `scipy.integrate.odeint` by extending the `autograd` automatic differentiation package. This code also supports all higher-order derivatives. We have since released a PyTorch (Paszke et al., 2017) implementation, including GPU-based implementations of several standard ODE solvers at `github.com/rtqichen/torchdiffeq`.

## 3   Replacing residual networks with ODEs for supervised learning

In this section, we experimentally investigate the training of neural ODEs for supervised learning.

**Software**   To solve ODE initial value problems numerically, we use the implicit Adams method implemented in LSODE and VODE and interfaced through the `scipy.integrate` package. Being an implicit method, it has better guarantees than explicit methods such as Runge-Kutta but requires solving a nonlinear optimization problem at every step. This setup makes direct backpropagation through the integrator difficult. We implement the adjoint sensitivity method in Python's `autograd` framework (Maclaurin et al., 2015). For the experiments in this section, we evaluated the hidden state dynamics and their derivatives on the GPU using Tensorflow, which were then called from the Fortran ODE solvers, which were called from Python `autograd` code.

**Model Architectures**   We experiment with a small residual network which downsamples the input twice then applies 6 standard residual blocks He et al. (2016b), which are replaced by an ODESolve module in the ODE-Net variant. We also test a network with the same architecture but where gradients are backpropagated directly through a Runge-Kutta integrator, re-

Table 1: Performance on MNIST. [†]From LeCun et al. (1998).

| | Test Error | # Params | Memory | Time |
|---|---|---|---|---|
| 1-Layer MLP[†] | 1.60% | 0.24 M | - | - |
| ResNet | 0.41% | 0.60 M | $\mathcal{O}(L)$ | $\mathcal{O}(L)$ |
| RK-Net | 0.47% | 0.22 M | $\mathcal{O}(\tilde{L})$ | $\mathcal{O}(\tilde{L})$ |
| ODE-Net | 0.42% | 0.22 M | $\mathcal{O}(1)$ | $\mathcal{O}(\tilde{L})$ |

ferred to as RK-Net. Table 1 shows test error, number of parameters, and memory cost. $L$ denotes the number of layers in the ResNet, and $\tilde{L}$ is the number of function evaluations that the ODE solver requests in a single forward pass, which can be interpreted as an implicit number of layers.

We find that ODE-Nets and RK-Nets can achieve around the same performance as the ResNet, while using fewer parameters. For reference, a neural net with a single hidden layer of 300 units has around the same number of parameters as the ODE-Net and RK-Net architecture that we tested.

**Error Control in ODE-Nets**    ODE solvers can approximately ensure that the output is within a given tolerance of the true solution. Changing this tolerance changes the behavior of the network. We first verify that error can indeed be controlled in Figure 3a. The time spent by the forward call is proportional to the number of function evaluations (Figure 3b), so tuning the tolerance gives us a trade-off between accuracy and computational cost. One could train with high accuracy, but switch to a lower accuracy at test time.

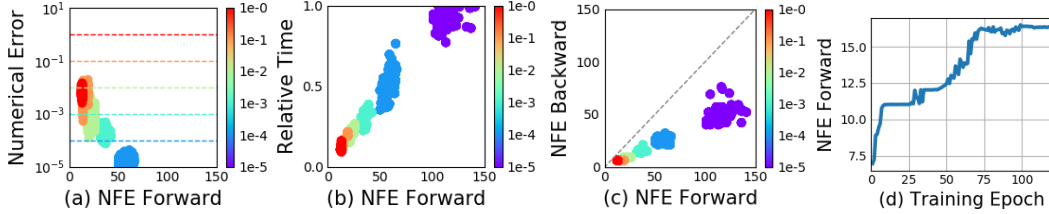

Figure 3: Statistics of a trained ODE-Net. (NFE = number of function evaluations.)

Figure 3c) shows a surprising result: the number of evaluations in the backward pass is roughly half of the forward pass. This suggests that the adjoint sensitivity method is not only more memory efficient, but also more computationally efficient than directly backpropagating through the integrator, because the latter approach will need to backprop through each function evaluation in the forward pass.

**Network Depth**    It's not clear how to define the 'depth' of an ODE solution. A related quantity is the number of evaluations of the hidden state dynamics required, a detail delegated to the ODE solver and dependent on the initial state or input. Figure 3d shows that he number of function evaluations increases throughout training, presumably adapting to increasing complexity of the model.

## 4    Continuous Normalizing Flows

The discretized equation (1) also appears in normalizing flows (Rezende and Mohamed, 2015) and the NICE framework (Dinh et al., 2014). These methods use the change of variables theorem to compute exact changes in probability if samples are transformed through a bijective function $f$:

$$\mathbf{z}_1 = f(\mathbf{z}_0) \implies \log p(\mathbf{z}_1) = \log p(\mathbf{z}_0) - \log \left| \det \frac{\partial f}{\partial \mathbf{z}_0} \right| \tag{6}$$

An example is the planar normalizing flow (Rezende and Mohamed, 2015):

$$\mathbf{z}(t + 1) = \mathbf{z}(t) + uh(w^\mathsf{T}\mathbf{z}(t) + b), \quad \log p(\mathbf{z}(t + 1)) = \log p(\mathbf{z}(t)) - \log \left| 1 + u^\mathsf{T} \frac{\partial h}{\partial \mathbf{z}} \right| \tag{7}$$

Generally, the main bottleneck to using the change of variables formula is computing of the determinant of the Jacobian $\partial f/\partial \mathbf{z}$, which has a cubic cost in either the dimension of $\mathbf{z}$, or the number of hidden units. Recent work explores the tradeoff between the expressiveness of normalizing flow layers and computational cost (Kingma et al., 2016; Tomczak and Welling, 2016; Berg et al., 2018).

Surprisingly, moving from a discrete set of layers to a continuous transformation simplifies the computation of the change in normalizing constant:

**Theorem 1** (Instantaneous Change of Variables). *Let* $\mathbf{z}(t)$ *be a finite continuous random variable with probability* $p(\mathbf{z}(t))$ *dependent on time. Let* $\frac{d\mathbf{z}}{dt} = f(\mathbf{z}(t), t)$ *be a differential equation describing a continuous-in-time transformation of* $\mathbf{z}(t)$*. Assuming that* $f$ *is uniformly Lipschitz continuous in* $\mathbf{z}$ *and continuous in* $t$*, then the change in log probability also follows a differential equation,*

$$\frac{\partial \log p(\mathbf{z}(t))}{\partial t} = -\mathrm{tr}\left( \frac{df}{d\mathbf{z}(t)} \right) \tag{8}$$

Proof in Appendix A. Instead of the log determinant in (6), we now only require a trace operation. Also unlike standard finite flows, the differential equation $f$ does not need to be bijective, since if uniqueness is satisfied, then the entire transformation is automatically bijective.

As an example application of the instantaneous change of variables, we can examine the continuous analog of the planar flow, and its change in normalization constant:

$$\frac{d\mathbf{z}(t)}{dt} = uh(w^{\mathsf{T}}\mathbf{z}(t) + b), \quad \frac{\partial \log p(\mathbf{z}(t))}{\partial t} = -u^{\mathsf{T}}\frac{\partial h}{\partial \mathbf{z}(t)} \quad (9)$$

Given an initial distribution $p(\mathbf{z}(0))$, we can sample from $p(\mathbf{z}(t))$ and evaluate its density by solving this combined ODE.

**Using multiple hidden units with linear cost** While det is not a linear function, the trace function is, which implies $\mathrm{tr}(\sum_n J_n) = \sum_n \mathrm{tr}(J_n)$. Thus if our dynamics is given by a sum of functions then the differential equation for the log density is also a sum:

$$\frac{d\mathbf{z}(t)}{dt} = \sum_{n=1}^{M} f_n(\mathbf{z}(t)), \quad \frac{d\log p(\mathbf{z}(t))}{dt} = \sum_{n=1}^{M} \mathrm{tr}\left(\frac{\partial f_n}{\partial \mathbf{z}}\right) \quad (10)$$

This means we can cheaply evaluate flow models having many hidden units, with a cost only linear in the number of hidden units $M$. Evaluating such 'wide' flow layers using standard normalizing flows costs $\mathcal{O}(M^3)$, meaning that standard NF architectures use many layers of only a single hidden unit.

**Time-dependent dynamics** We can specify the parameters of a flow as a function of $t$, making the differential equation $f(\mathbf{z}(t), t)$ change with $t$. This is parameterization is a kind of hypernetwork (Ha et al., 2016). We also introduce a gating mechanism for each hidden unit, $\frac{d\mathbf{z}}{dt} = \sum_n \sigma_n(t) f_n(\mathbf{z})$ where $\sigma_n(t) \in (0, 1)$ is a neural network that learns when the dynamic $f_n(\mathbf{z})$ should be applied. We call these models continuous normalizing flows (CNF).

## 4.1 Experiments with Continuous Normalizing Flows

We first compare continuous and discrete planar flows at learning to sample from a known distribution. We show that a planar CNF with $M$ hidden units can be at least as expressive as a planar NF with $K = M$ layers, and sometimes much more expressive.

**Density matching** We configure the CNF as described above, and train for 10,000 iterations using Adam (Kingma and Ba, 2014). In contrast, the NF is trained for 500,000 iterations using RMSprop (Hinton et al., 2012), as suggested by Rezende and Mohamed (2015). For this task, we minimize KL $(q(\mathbf{x})\|p(\mathbf{x}))$ as the loss function where $q$ is the flow model and the target density $p(\cdot)$ can be evaluated. Figure 4 shows that CNF generally achieves lower loss.

**Maximum Likelihood Training** A useful property of continuous-time normalizing flows is that we can compute the reverse transformation for about the same cost as the forward pass, which cannot be said for normalizing flows. This lets us train the flow on a density estimation task by performing

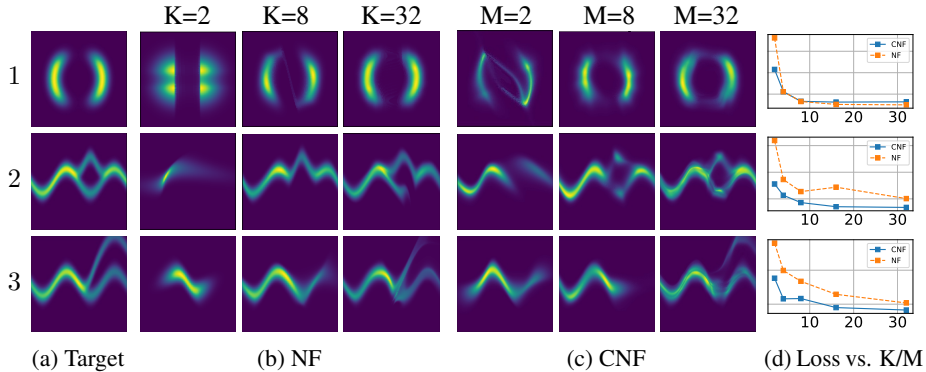

Figure 4: Comparison of normalizing flows versus continuous normalizing flows. The model capacity of normalizing flows is determined by their depth (K), while continuous normalizing flows can also increase capacity by increasing width (M), making them easier to train.

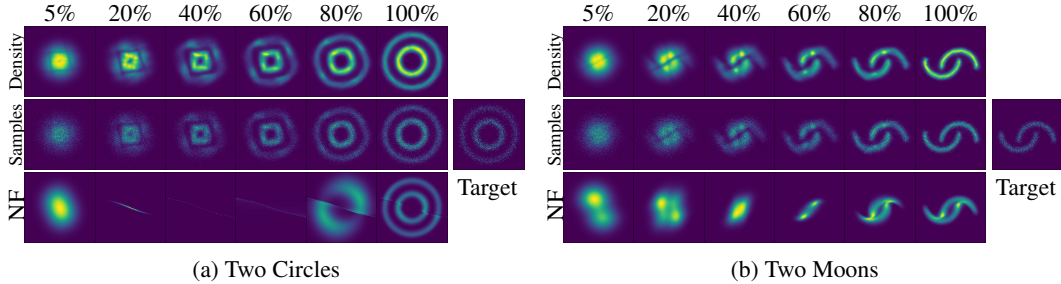

|  | 5% | 20% | 40% | 60% | 80% | 100% |
| Density | | | | | | |
| Samples | | | | | | |
| NF | | | | | | |

(a) Two Circles

(b) Two Moons

Figure 5: **Visualizing the transformation from noise to data.** Continuous-time normalizing flows are reversible, so we can train on a density estimation task and still be able to sample from the learned density efficiently.

maximum likelihood estimation, which maximizes $\mathbb{E}_{p(\mathbf{x})}[\log q(\mathbf{x})]$ where $q(\cdot)$ is computed using the appropriate change of variables theorem, then afterwards reverse the CNF to generate random samples from $q(\mathbf{x})$.

For this task, we use 64 hidden units for CNF, and 64 stacked one-hidden-unit layers for NF. Figure 5 shows the learned dynamics. Instead of showing the initial Gaussian distribution, we display the transformed distribution after a small amount of time which shows the locations of the initial planar flows. Interestingly, to fit the Two Circles distribution, the CNF rotates the planar flows so that the particles can be evenly spread into circles. While the CNF transformations are smooth and interpretable, we find that NF transformations are very unintuitive and this model has difficulty fitting the two moons dataset in Figure 5b.

## 5 A generative latent function time-series model

Applying neural networks to irregularly-sampled data such as medical records, network traffic, or neural spiking data is difficult. Typically, observations are put into bins of fixed duration, and the latent dynamics are discretized in the same way. This leads to difficulties with missing data and ill-defined latent variables. Missing data can be addressed using generative time-series models (Álvarez and Lawrence, 2011; Futoma et al., 2017; Mei and Eisner, 2017; Soleimani et al., 2017a) or data imputation (Che et al., 2018). Another approach concatenates time-stamp information to the input of an RNN (Choi et al., 2016; Lipton et al., 2016; Du et al., 2016; Li, 2017).

We present a continuous-time, generative approach to modeling time series. Our model represents each time series by a latent trajectory. Each trajectory is determined from a local initial state, $\mathbf{z}_{t_0}$, and a global set of latent dynamics shared across all time series. Given observation times $t_0, t_1, \ldots, t_N$ and an initial state $\mathbf{z}_{t_0}$, an ODE solver produces $\mathbf{z}_{t_1}, \ldots, \mathbf{z}_{t_N}$, which describe the latent state at each observation. We define this generative model formally through a sampling procedure:

$$\mathbf{z}_{t_0} \sim p(\mathbf{z}_{t_0}) \tag{11}$$
$$\mathbf{z}_{t_1}, \mathbf{z}_{t_2}, \ldots, \mathbf{z}_{t_N} = \text{ODESolve}(\mathbf{z}_{t_0}, f, \theta_f, t_0, \ldots, t_N) \tag{12}$$
$$\text{each} \quad \mathbf{x}_{t_i} \sim p(\mathbf{x}|\mathbf{z}_{t_i}, \theta_\mathbf{x}) \tag{13}$$

Function $f$ is a time-invariant function that takes the value $\mathbf{z}$ at the current time step and outputs the gradient: $\partial \mathbf{z}(t)/\partial t = f(\mathbf{z}(t), \theta_f)$. We parametrize this function using a neural net. Because $f$ is time-invariant, given any latent state $\mathbf{z}(t)$, the entire latent trajectory is uniquely defined. Extrapolating this latent trajectory lets us make predictions arbitrarily far forwards or backwards in time.

**Training and Prediction** We can train this latent-variable model as a variational autoencoder (Kingma and Welling, 2014; Rezende et al., 2014), with sequence-valued observations. Our recognition net is an RNN, which consumes the data sequentially backwards in time, and outputs $q_\phi(\mathbf{z}_0|\mathbf{x}_1, \mathbf{x}_2, \ldots, \mathbf{x}_N)$. A detailed algorithm can be found in Appendix E. Using ODEs as a generative model allows us to make predictions for arbitrary time points $t_1...t_M$ on a continuous timeline.

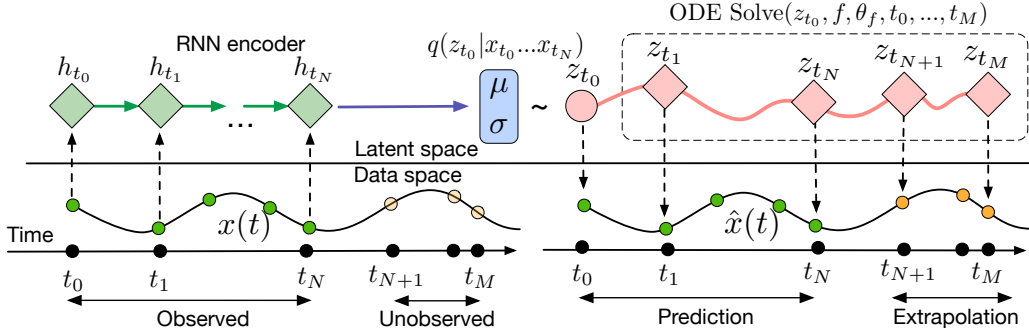

Figure 6: Computation graph of the latent ODE model.

**Poisson Process likelihoods** The fact that an observation occurred often tells us something about the latent state. For example, a patient may be more likely to take a medical test if they are sick. The rate of events can be parameterized by a function of the latent state: $p(\text{event at time } t \,|\, \mathbf{z}(t)) = \lambda(\mathbf{z}(t))$. Given this rate function, the likelihood of a set of independent observation times in the interval $[t_{\text{start}}, t_{\text{end}}]$ is given by an inhomogeneous Poisson process (Palm, 1943):

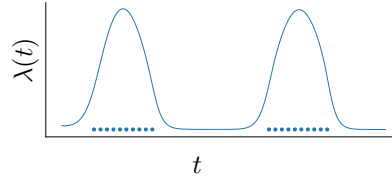

Figure 7: Fitting a latent ODE dynamics model with a Poisson process likelihood. Dots show event times. The line is the learned intensity $\lambda(t)$ of the Poisson process.

$$\log p(t_1 \ldots t_N \,|\, t_{\text{start}}, t_{\text{end}}) = \sum_{i=1}^{N} \log \lambda(\mathbf{z}(t_i)) - \int_{t_{\text{start}}}^{t_{\text{end}}} \lambda(\mathbf{z}(t))dt$$

We can parameterize $\lambda(\cdot)$ using another neural network. Conveniently, we can evaluate both the latent trajectory and the Poisson process likelihood together in a single call to an ODE solver. Figure 7 shows the event rate learned by such a model on a toy dataset.

A Poisson process likelihood on observation times can be combined with a data likelihood to jointly model all observations and the times at which they were made.

## 5.1 Time-series Latent ODE Experiments

We investigate the ability of the latent ODE model to fit and extrapolate time series. The recognition network is an RNN with 25 hidden units. We use a 4-dimensional latent space. We parameterize the dynamics function $f$ with a one-hidden-layer network with 20 hidden units. The decoder computing $p(\mathbf{x}_{t_i}|\mathbf{z}_{t_i})$ is another neural network with one hidden layer with 20 hidden units. Our baseline was a recurrent neural net with 25 hidden units trained to minimize negative Gaussian log-likelihood. We trained a second version of this RNN whose inputs were concatenated with the time difference to the next observation to aid RNN with irregular observations.

**Bi-directional spiral dataset** We generated a dataset of 1000 2-dimensional spirals, each starting at a different point, sampled at 100 equally-spaced timesteps. The dataset contains two types of spirals: half are clockwise while the other half counter-clockwise. To make the task more realistic, we add gaussian noise to the observations.

Table 2: Predictive RMSE on test set

| # Observations | 30/100 | 50/100 | 100/100 |
|---|---|---|---|
| RNN | 0.3937 | 0.3202 | 0.1813 |
| Latent ODE | **0.1642** | **0.1502** | **0.1346** |

**Time series with irregular time points** To generate irregular timestamps, we randomly sample points from each trajectory without replacement ($n = \{30, 50, 100\}$). We report predictive root-mean-squared error (RMSE) on 100 time points extending beyond those that were used for training. Table 2 shows that the latent ODE has substantially lower predictive RMSE.

Figure 8 shows examples of spiral reconstructions with 30 sub-sampled points. Reconstructions from the latent ODE were obtained by sampling from the posterior over latent trajectories and decoding it

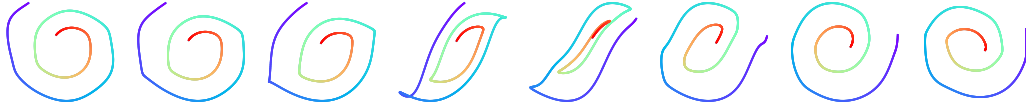

Figure 9: Data-space trajectories decoded from varying one dimension of $\mathbf{z}_{t_0}$. Color indicates progression through time, starting at purple and ending at red. Note that the trajectories on the left are counter-clockwise, while the trajectories on the right are clockwise.

to data-space. Examples with varying number of time points are shown in Appendix F. We observed that reconstructions and extrapolations are consistent with the ground truth regardless of number of observed points and despite the noise.

**Latent space interpolation** Figure 8c shows latent trajectories projected onto the first two dimensions of the latent space. The trajectories form two separate clusters of trajectories, one decoding to clockwise spirals, the other to counter-clockwise. Figure 9 shows that the latent trajectories change smoothly as a function of the initial point $\mathbf{z}(t_0)$, switching from a clockwise to a counter-clockwise spiral.

## 6 Scope and Limitations

**Minibatching** The use of mini-batches is less straightforward than for standard neural networks. One can still batch together evaluations through the ODE solver by concatenating the states of each batch element together, creating a combined ODE with dimension $D \times K$. In some cases, controlling error on all batch elements together might require evaluating the combined system $K$ times more often than if each system was solved individually. However, in practice the number of evaluations did not increase substantially when using minibatches.

**Uniqueness** When do continuous dynamics have a unique solution? Picard's existence theorem (Coddington and Levinson, 1955) states that the solution to an initial value problem exists and is unique if the differential equation is uniformly Lipschitz continuous in $\mathbf{z}$ and continuous in $t$. This theorem holds for our model if the neural network has finite weights and uses Lipshitz nonlinearities, such as `tanh` or `relu`.

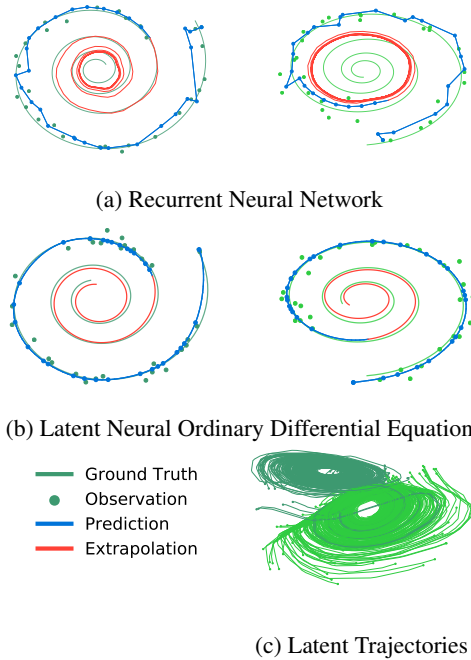

(a) Recurrent Neural Network

(b) Latent Neural Ordinary Differential Equation

- Ground Truth
- Observation
- Prediction
- Extrapolation

(c) Latent Trajectories

Figure 8: (a): Reconstruction and extrapolation of spirals with irregular time points by a recurrent neural network. (b): Reconstructions and extrapolations by a latent neural ODE. Blue curve shows model prediction. Red shows extrapolation. (c) A projection of inferred 4-dimensional latent ODE trajectories onto their first two dimensions. Color indicates the direction of the corresponding trajectory. The model has learned latent dynamics which distinguishes the two directions.

**Setting tolerances** Our framework allows the user to trade off speed for precision, but requires the user to choose an error tolerance on both the forward and reverse passes during training. For sequence modeling, the default value of `1.5e-8` was used. In the classification and density estimation experiments, we were able to reduce the tolerance to `1e-3` and `1e-5`, respectively, without degrading performance.

**Reconstructing forward trajectories** Reconstructing the state trajectory by running the dynamics backwards can introduce extra numerical error if the reconstructed trajectory diverges from the original. This problem can be addressed by checkpointing: storing intermediate values of $\mathbf{z}$ on the

forward pass, and reconstructing the exact forward trajectory by re-integrating from those points. We did not find this to be a practical problem, and we informally checked that reversing many layers of continuous normalizing flows with default tolerances recovered the initial states.

# 7    Related Work

The use of the adjoint method for training continuous-time neural networks was previously proposed (LeCun et al., 1988; Pearlmutter, 1995), though was not demonstrated practically. The interpretation of residual networks He et al. (2016a) as approximate ODE solvers spurred research into exploiting reversibility and approximate computation in ResNets (Chang et al., 2017; Lu et al., 2017). We demonstrate these same properties in more generality by directly using an ODE solver.

**Adaptive computation**    One can adapt computation time by training secondary neural networks to choose the number of evaluations of recurrent or residual networks (Graves, 2016; Jernite et al., 2016; Figurnov et al., 2017; Chang et al., 2018). However, this introduces overhead both at training and test time, and extra parameters that need to be fit. In contrast, ODE solvers offer well-studied, computationally cheap, and generalizable rules for adapting the amount of computation.

**Constant memory backprop through reversibility**    Recent work developed reversible versions of residual networks (Gomez et al., 2017; Haber and Ruthotto, 2017; Chang et al., 2017), which gives the same constant memory advantage as our approach. However, these methods require restricted architectures, which partition the hidden units. Our approach does not have these restrictions.

**Learning differential equations**    Much recent work has proposed learning differential equations from data. One can train feed-forward or recurrent neural networks to approximate a differential equation (Raissi and Karniadakis, 2018; Raissi et al., 2018a; Long et al., 2017), with applications such as fluid simulation (Wiewel et al., 2018). There is also significant work on connecting Gaussian Processes (GPs) and ODE solvers (Schober et al., 2014). GPs have been adapted to fit differential equations (Raissi et al., 2018b) and can naturally model continuous-time effects and interventions (Soleimani et al., 2017b; Schulam and Saria, 2017). Ryder et al. (2018) use stochastic variational inference to recover the solution of a given stochastic differential equation.

**Differentiating through ODE solvers**    The `dolfin` library (Farrell et al., 2013) implements adjoint computation for general ODE and PDE solutions, but only by backpropagating through the individual operations of the forward solver. The Stan library (Carpenter et al., 2015) implements gradient estimation through ODE solutions using forward sensitivity analysis. However, forward sensitivity analysis is quadratic-time in the number of variables, whereas the adjoint sensitivity analysis is linear (Carpenter et al., 2015; Zhang and Sandu, 2014). Melicher et al. (2017) used the adjoint method to train bespoke latent dynamic models.

In contrast, by providing a generic vector-Jacobian product, we allow an ODE solver to be trained end-to-end with any other differentiable model components. While use of vector-Jacobian products for solving the adjoint method has been explored in optimal control (Andersson, 2013; Andersson et al., In Press, 2018), we highlight the potential of a general integration of black-box ODE solvers into automatic differentiation (Baydin et al., 2018) for deep learning and generative modeling.

# 8    Conclusion

We investigated the use of black-box ODE solvers as a model component, developing new models for time-series modeling, supervised learning, and density estimation. These models are evaluated adaptively, and allow explicit control of the tradeoff between computation speed and accuracy. Finally, we derived an instantaneous version of the change of variables formula, and developed continuous-time normalizing flows, which can scale to large layer sizes.

# 9 Acknowledgements

We thank Wenyi Wang and Geoff Roeder for help with proofs, and Daniel Duckworth, Ethan Fetaya, Hossein Soleimani, Eldad Haber, Ken Caluwaerts, and Daniel Flam-Shepherd for feedback. We thank Chris Rackauckas, Dougal Maclaurin, and Matthew James Johnson for helpful discussions.

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
