[Supplementary Material]

## Appendix A    Proof of the Instantaneous Change of Variables Theorem

**Theorem** (Instantaneous Change of Variables). *Let $\mathbf{z}(t)$ be a finite continuous random variable with probability $p(\mathbf{z}(t))$ dependent on time. Let $\frac{d\mathbf{z}}{dt} = f(\mathbf{z}(t), t)$ be a differential equation describing a continuous-in-time transformation of $\mathbf{z}(t)$. Assuming that $f$ is uniformly Lipschitz continuous in $\mathbf{z}$ and continuous in $t$, then the change in log probability also follows a differential equation:*

$$\frac{\partial \log p(\mathbf{z}(t))}{\partial t} = -\text{tr}\left(\frac{df}{d\mathbf{z}}(t)\right)$$

*Proof.* To prove this theorem, we take the infinitesimal limit of finite changes of $\log p(\mathbf{z}(t))$ through time. First we denote the transformation of $\mathbf{z}$ over an $\varepsilon$ change in time as

$$\mathbf{z}(t + \varepsilon) = T_\varepsilon(\mathbf{z}(t)) \tag{14}$$

We assume that $f$ is Lipschitz continuous in $\mathbf{z}(t)$ and continuous in $t$, so every initial value problem has a unique solution by Picard's existence theorem. We also assume $\mathbf{z}(t)$ is bounded. These conditions imply that $f$, $T_\varepsilon$, and $\frac{\partial}{\partial \mathbf{z}} T_\varepsilon$ are all bounded. In the following, we use these conditions to exchange limits and products.

We can write the differential equation $\frac{\partial \log p(\mathbf{z}(t))}{\partial t}$ using the discrete change of variables formula, and the definition of the derivative:

$$\frac{\partial \log p(\mathbf{z}(t))}{\partial t} = \lim_{\varepsilon \to 0^+} \frac{\log p(\mathbf{z}(t)) - \log \left|\det \frac{\partial}{\partial \mathbf{z}} T_\varepsilon(\mathbf{z}(t))\right| - \log p(\mathbf{z}(t))}{\varepsilon} \tag{15}$$

$$= -\lim_{\varepsilon \to 0^+} \frac{\log \left|\det \frac{\partial}{\partial \mathbf{z}} T_\varepsilon(\mathbf{z}(t))\right|}{\varepsilon} \tag{16}$$

$$= -\lim_{\varepsilon \to 0^+} \frac{\frac{\partial}{\partial \varepsilon} \log \left|\det \frac{\partial}{\partial \mathbf{z}} T_\varepsilon(\mathbf{z}(t))\right|}{\frac{\partial}{\partial \varepsilon} \varepsilon} \qquad \text{(by L'Hôpital's rule)} \tag{17}$$

$$= -\lim_{\varepsilon \to 0^+} \frac{\frac{\partial}{\partial \varepsilon} \left|\det \frac{\partial}{\partial \mathbf{z}} T_\varepsilon(\mathbf{z}(t))\right|}{\left|\det \frac{\partial}{\partial \mathbf{z}} T_\varepsilon(\mathbf{z}(t))\right|} \qquad \left(\left.\frac{\partial \log(\mathbf{z})}{\partial \mathbf{z}}\right|_{\mathbf{z}=1} = 1\right) \tag{18}$$

$$= -\underbrace{\left(\lim_{\varepsilon \to 0^+} \frac{\partial}{\partial \varepsilon} \left|\det \frac{\partial}{\partial \mathbf{z}} T_\varepsilon(\mathbf{z}(t))\right|\right)}_{\text{bounded}} \underbrace{\left(\lim_{\varepsilon \to 0^+} \frac{1}{\left|\det \frac{\partial}{\partial \mathbf{z}} T_\varepsilon(\mathbf{z}(t))\right|}\right)}_{=1} \tag{19}$$

$$= -\lim_{\varepsilon \to 0^+} \frac{\partial}{\partial \varepsilon} \left|\det \frac{\partial}{\partial \mathbf{z}} T_\varepsilon(\mathbf{z}(t))\right| \tag{20}$$

The derivative of the determinant can be expressed using Jacobi's formula, which gives

$$\frac{\partial \log p(\mathbf{z}(t))}{\partial t} = -\lim_{\varepsilon \to 0^+} \text{tr}\left(\text{adj}\left(\frac{\partial}{\partial \mathbf{z}} T_\varepsilon(\mathbf{z}(t))\right) \frac{\partial}{\partial \varepsilon} \frac{\partial}{\partial \mathbf{z}} T_\varepsilon(\mathbf{z}(t))\right) \tag{21}$$

$$= -\text{tr}\left(\underbrace{\left(\lim_{\varepsilon \to 0^+} \text{adj}\left(\frac{\partial}{\partial \mathbf{z}} T_\varepsilon(\mathbf{z}(t))\right)\right)}_{=I} \left(\lim_{\varepsilon \to 0^+} \frac{\partial}{\partial \varepsilon} \frac{\partial}{\partial \mathbf{z}} T_\varepsilon(\mathbf{z}(t))\right)\right) \tag{22}$$

$$= -\text{tr}\left(\lim_{\varepsilon \to 0^+} \frac{\partial}{\partial \varepsilon} \frac{\partial}{\partial \mathbf{z}} T_\varepsilon(\mathbf{z}(t))\right) \tag{23}$$

Substituting $T_\varepsilon$ with its Taylor series expansion and taking the limit, we complete the proof.

$$\frac{\partial \log p(\mathbf{z}(t))}{\partial t} = -\text{tr}\left(\lim_{\varepsilon \to 0^+} \frac{\partial}{\partial \varepsilon} \frac{\partial}{\partial \mathbf{z}} \left(\mathbf{z} + \varepsilon f(\mathbf{z}(t), t) + \mathcal{O}(\varepsilon^2) + \mathcal{O}(\varepsilon^3) + \dots\right)\right) \tag{24}$$

$$= -\text{tr}\left(\lim_{\varepsilon \to 0^+} \frac{\partial}{\partial \varepsilon} \left(I + \frac{\partial}{\partial \mathbf{z}} \varepsilon f(\mathbf{z}(t), t) + \mathcal{O}(\varepsilon^2) + \mathcal{O}(\varepsilon^3) + \dots\right)\right) \tag{25}$$

$$= -\text{tr}\left(\lim_{\varepsilon \to 0^+} \left(\frac{\partial}{\partial \mathbf{z}} f(\mathbf{z}(t), t) + \mathcal{O}(\varepsilon) + \mathcal{O}(\varepsilon^2) + \dots\right)\right) \tag{26}$$

$$= -\text{tr}\left(\frac{\partial}{\partial \mathbf{z}} f(\mathbf{z}(t), t)\right) \tag{27}$$

$\square$

### A.1 Special Cases

**Planar CNF.** Let $f(\mathbf{z}) = uh(w^{\mathbf{z}} + b)$, then $\frac{\partial f}{\partial \mathbf{z}} = u \frac{\partial h}{\partial \mathbf{z}}^{\mathsf{T}}$. Since the trace of an outer product is the inner product, we have

$$\frac{\partial \log p(\mathbf{z})}{\partial t} = -\mathrm{tr}\left(u \frac{\partial h}{\partial \mathbf{z}}^{\mathsf{T}}\right) = -u^{\mathsf{T}} \frac{\partial h}{\partial \mathbf{z}} \tag{28}$$

This is the parameterization we use in all of our experiments.

**Hamiltonian CNF.** The continuous analog of NICE (Dinh et al., 2014) is a Hamiltonian flow, which splits the data into two equal partitions and is a volume-preserving transformation, implying that $\frac{\partial \log p(\mathbf{z})}{\partial t} = 0$. We can verify this. Let

$$\begin{bmatrix} \frac{d\mathbf{z}_{1:d}}{dt} \\ \frac{d\mathbf{z}_{d+1:D}}{dt} \end{bmatrix} = \begin{bmatrix} f(\mathbf{z}_{d+1:D}) \\ g(\mathbf{z}_{1:d}) \end{bmatrix} \tag{29}$$

Then because the Jacobian is all zeros on its diagonal, the trace is zero. This is a volume-preserving flow.

### A.2 Connection to Fokker-Planck and Liouville PDEs

The Fokker-Planck equation is a well-known partial differential equation (PDE) that describes the probability density function of a stochastic differential equation as it changes with time. We relate the instantaneous change of variables to the special case of Fokker-Planck with zero diffusion, the Liouville equation.

As with the instantaneous change of variables, let $\mathbf{z}(t) \in \mathbb{R}^D$ evolve through time following $\frac{d\mathbf{z}(t)}{dt} = f(\mathbf{z}(t), t)$. Then Liouville equation describes the change in density of $\mathbf{z}$–*a fixed point in space*–as a PDE,

$$\frac{\partial p(\mathbf{z}, t)}{\partial t} = -\sum_{i=1}^{D} \frac{\partial}{\partial \mathbf{z}_i} \left[ f_i(\mathbf{z}, t) p(\mathbf{z}, t) \right] \tag{30}$$

However, (30) cannot be easily used as it requires the partial derivatives of $\frac{p(\mathbf{z},t)}{\partial \mathbf{z}}$, which is typically approximated using finite difference. This type of PDE has its own literature on efficient and accurate simulation (Stam, 1999).

Instead of evaluating $p(\cdot, t)$ at a fixed point, if we follow the trajectory of a particle $\mathbf{z}(t)$, we obtain

$$\frac{\partial p(\mathbf{z}(t), t)}{\partial t} = \underbrace{\frac{\partial p(\mathbf{z}(t), t)}{\partial \mathbf{z}(t)} \frac{\partial \mathbf{z}(t)}{\partial t}}_{\text{partial derivative from first argument, } \mathbf{z}(t)} + \underbrace{\frac{\partial p(\mathbf{z}(t), t)}{\partial t}}_{\text{partial derivative from second argument, } t}$$

$$= \sum_{i=1}^{D} \frac{\partial p(\mathbf{z}(t), t)}{\partial \mathbf{z}_i(t)} \frac{\partial \mathbf{z}_i(t)}{\partial t} - \sum_{i=1}^{D} \frac{\partial f_i(\mathbf{z}(t), t)}{\partial \mathbf{z}_i} p(\mathbf{z}(t), t) - \sum_{i=1}^{D} f_i(\mathbf{z}(t), t) \frac{\partial p(\mathbf{z}(t), t)}{\partial \mathbf{z}_i(t)} \tag{31}$$

$$= -\sum_{i=1}^{D} \frac{\partial f_i(\mathbf{z}(t), t)}{\partial \mathbf{z}_i} p(\mathbf{z}(t), t)$$

We arrive at the instantaneous change of variables by taking the log,

$$\frac{\partial \log p(\mathbf{z}(t), t)}{\partial t} = \frac{1}{p(\mathbf{z}(t), t)} \frac{\partial p(\mathbf{z}(t), t)}{\partial t} = -\sum_{i=1}^{D} \frac{\partial f_i(\mathbf{z}(t), t)}{\partial \mathbf{z}_i} \tag{32}$$

While still a PDE, (32) can be combined with $\mathbf{z}(t)$ to form an ODE of size $D + 1$,

$$\frac{d}{dt} \begin{bmatrix} \mathbf{z}(t) \\ \log p(\mathbf{z}(t), t) \end{bmatrix} = \begin{bmatrix} f(\mathbf{z}(t), t) \\ -\sum_{i=1}^{D} \frac{\partial f_i(\mathbf{z}(t), t)}{\partial t} \end{bmatrix} \tag{33}$$

Compared to the Fokker-Planck and Liouville equations, the instantaneous change of variables is of more practical impact as it can be numerically solved much more easily, requiring an extra state of $D$ for following the trajectory of $\mathbf{z}(t)$. Whereas an approach based on finite difference approximation of the Liouville equation would require a grid size that is exponential in $D$.

## Appendix B    A Modern Proof of the Adjoint Method

We present an alternative proof to the adjoint method (Pontryagin et al., 1962) that is short and easy to follow.

## B.1  Continuous Backpropagation

Let $\mathbf{z}(t)$ follow the differential equation $\frac{d\mathbf{z}(t)}{dt} = f(\mathbf{z}(t), t, \theta)$, where $\theta$ are the parameters. We will prove that if we define an adjoint state

$$\mathbf{a}(t) = \frac{dL}{d\mathbf{z}(t)} \tag{34}$$

then it follows the differential equation

$$\frac{d\mathbf{a}(t)}{dt} = -\mathbf{a}(t)\frac{\partial f(\mathbf{z}(t), t, \theta)}{\partial \mathbf{z}(t)} \tag{35}$$

For ease of notation, we denote vectors as row vectors, whereas the main text uses column vectors.

The adjoint state is the gradient with respect to the hidden state at a specified time $t$. In standard neural networks, the gradient of a hidden layer $\mathbf{h}_t$ depends on the gradient from the next layer $\mathbf{h}_{t+1}$ by chain rule

$$\frac{dL}{d\mathbf{h}_t} = \frac{dL}{d\mathbf{h}_{t+1}}\frac{d\mathbf{h}_{t+1}}{d\mathbf{h}_t}. \tag{36}$$

With a continuous hidden state, we can write the transformation after an $\varepsilon$ change in time as

$$\mathbf{z}(t+\varepsilon) = \int_t^{t+\varepsilon} f(\mathbf{z}(t), t, \theta)dt + \mathbf{z}(t) = T_\varepsilon(\mathbf{z}(t), t) \tag{37}$$

and chain rule can also be applied

$$\frac{dL}{\partial \mathbf{z}(t)} = \frac{dL}{d\mathbf{z}(t+\varepsilon)}\frac{d\mathbf{z}(t+\varepsilon)}{d\mathbf{z}(t)} \qquad \text{or} \qquad \mathbf{a}(t) = \mathbf{a}(t+\varepsilon)\frac{\partial T_\varepsilon(\mathbf{z}(t), t)}{\partial \mathbf{z}(t)} \tag{38}$$

The proof of (35) follows from the definition of derivative:

$$\frac{d\mathbf{a}(t)}{dt} = \lim_{\varepsilon \to 0^+} \frac{\mathbf{a}(t+\varepsilon) - \mathbf{a}(t)}{\varepsilon} \tag{39}$$

$$= \lim_{\varepsilon \to 0^+} \frac{\mathbf{a}(t+\varepsilon) - \mathbf{a}(t+\varepsilon)\frac{\partial}{\partial \mathbf{z}(t)}T_\varepsilon(\mathbf{z}(t))}{\varepsilon} \qquad \text{(by Eq 38)} \tag{40}$$

$$= \lim_{\varepsilon \to 0^+} \frac{\mathbf{a}(t+\varepsilon) - \mathbf{a}(t+\varepsilon)\frac{\partial}{\partial \mathbf{z}(t)}\left(\mathbf{z}(t) + \varepsilon f(\mathbf{z}(t), t, \theta) + \mathcal{O}(\varepsilon^2)\right)}{\varepsilon} \qquad \text{(Taylor series around } \mathbf{z}(t)) \tag{41}$$

$$= \lim_{\varepsilon \to 0^+} \frac{\mathbf{a}(t+\varepsilon) - \mathbf{a}(t+\varepsilon)\left(I + \varepsilon\frac{\partial f(\mathbf{z}(t), t, \theta)}{\partial \mathbf{z}(t)} + \mathcal{O}(\varepsilon^2)\right)}{\varepsilon} \tag{42}$$

$$= \lim_{\varepsilon \to 0^+} \frac{-\varepsilon \mathbf{a}(t+\varepsilon)\frac{\partial f(\mathbf{z}(t), t, \theta)}{\partial \mathbf{z}(t)} + \mathcal{O}(\varepsilon^2)}{\varepsilon} \tag{43}$$

$$= \lim_{\varepsilon \to 0^+} -\mathbf{a}(t+\varepsilon)\frac{\partial f(\mathbf{z}(t), t, \theta)}{\partial \mathbf{z}(t)} + \mathcal{O}(\varepsilon) \tag{44}$$

$$= -\mathbf{a}(t)\frac{\partial f(\mathbf{z}(t), t, \theta)}{\partial \mathbf{z}(t)} \tag{45}$$

We pointed out the similarity between adjoint method and backpropagation (eq. 38). Similarly to backpropagation, ODE for the adjoint state needs to be solved *backwards* in time. We specify the constraint on the last time point, which is simply the gradient of the loss wrt the last time point, and can obtain the gradients with respect to the hidden state at any time, including the initial value.

$$\underbrace{\mathbf{a}(t_N) = \frac{dL}{d\mathbf{z}(t_N)}}_{\text{initial condition of adjoint diffeq.}} \qquad \underbrace{\mathbf{a}(t_0) = \int_{t_N}^{t_0} \mathbf{a}(t)\frac{\partial f(\mathbf{z}(t), t, \theta)}{\partial \mathbf{z}(t)}\, dt}_{\text{gradient wrt. initial value}} \tag{46}$$

Here we assumed that loss function $L$ depends only on the last time point $t_N$. If function $L$ depends also on intermediate time points $t_1, t_2, \ldots, t_{N-1}$, etc., we can repeat the adjoint step for each of the intervals $[t_{N-1}, t_N]$, $[t_{N-2}, t_{N-1}]$ in the backward order and sum up the obtained gradients.

## B.2  Gradients wrt. $\theta$ and $t$

We can generalize (35) to obtain gradients with respect to $\theta$–a constant wrt. $t$–and and the initial and end times, $t_0$ and $t_N$. We view $\theta$ and $t$ as states with constant differential equations and write

$$\frac{\partial \theta(t)}{\partial t} = \mathbf{0} \qquad \frac{dt(t)}{dt} = 1 \tag{47}$$

We can then combine these with $z$ to form an augmented state[1] with corresponding differential equation and adjoint state,

$$\frac{d}{dt}\begin{bmatrix}\mathbf{z}\\ \theta\\ t\end{bmatrix}(t) = f_{aug}([\mathbf{z},\theta,t]) := \begin{bmatrix}f([\mathbf{z},\theta,t])\\ \mathbf{0}\\ 1\end{bmatrix},\ \mathbf{a}_{aug} := \begin{bmatrix}\mathbf{a}\\ \mathbf{a}_\theta\\ \mathbf{a}_t\end{bmatrix},\ \mathbf{a}_\theta(t) := \frac{dL}{d\theta(t)},\ \mathbf{a}_t(t) := \frac{dL}{dt(t)} \tag{48}$$

Note this formulates the augmented ODE as an autonomous (time-invariant) ODE, but the derivations in the previous section still hold as this is a special case of a time-variant ODE. The Jacobian of $f$ has the form

$$\frac{\partial f_{aug}}{\partial[\mathbf{z},\theta,t]} = \begin{bmatrix}\frac{\partial f}{\partial \mathbf{z}} & \frac{\partial f}{\partial \theta} & \frac{\partial f}{\partial t}\\ \mathbf{0} & \mathbf{0} & \mathbf{0}\\ \mathbf{0} & \mathbf{0} & \mathbf{0}\end{bmatrix}(t) \tag{49}$$

where each $\mathbf{0}$ is a matrix of zeros with the appropriate dimensions. We plug this into (35) to obtain

$$\frac{d\mathbf{a}_{aug}(t)}{dt} = -\begin{bmatrix}\mathbf{a}(t) & \mathbf{a}_\theta(t) & \mathbf{a}_t(t)\end{bmatrix}\frac{\partial f_{aug}}{\partial[\mathbf{z},\theta,t]}(t) = -\begin{bmatrix}\mathbf{a}\frac{\partial f}{\partial \mathbf{z}} & \mathbf{a}\frac{\partial f}{\partial \theta} & \mathbf{a}\frac{\partial f}{\partial t}\end{bmatrix}(t) \tag{50}$$

The first element is the adjoint differential equation (35), as expected. The second element can be used to obtain the total gradient with respect to the parameters, by integrating over the full interval.

$$\frac{dL}{d\theta} = \int_{t_N}^{t_0} \mathbf{a}(t)\frac{\partial f(\mathbf{z}(t),t,\theta)}{\partial \theta}\ dt \tag{51}$$

Note the negative sign cancels out since we integrate backwards from $t_N$ to $t_0$. Finally, we also get gradients with respect to $t_0$ and $t_N$, the start and end of the integration interval.

$$\frac{dL}{dt_N} = -\mathbf{a}(t_N)\frac{\partial f(\mathbf{z}(t_N),t_N,\theta)}{\partial t_N} \qquad \frac{dL}{dt_0} = \int_{t_N}^{t_0} \mathbf{a}(t)\frac{\partial f(\mathbf{z}(t),t,\theta)}{\partial t}\ dt \tag{52}$$

Between (35), (46), (51), and (52) we have gradients for all possible inputs to an initial value problem solver.

## Appendix C  Full Adjoint sensitivities algorithm

This more detailed version of Algorithm 1 includes gradients with respect to the start and end times of integration.

---
**Algorithm 2** Reverse-mode derivative of an ODE initial value problem

---
**Input:** dynamics parameters $\theta$, start time $t_0$, stop time $t_1$, final state $\mathbf{z}(t_1)$, loss gradient $\partial L/\partial \mathbf{z}(t_1)$

$\quad \frac{\partial L}{\partial t_1} = \frac{\partial L}{\partial \mathbf{z}(t_1)}^\mathsf{T} f(\mathbf{z}(t_1),t_1,\theta)$ $\qquad\qquad\qquad\qquad\qquad$ ▷ Compute gradient w.r.t. $t_1$

$\quad s_0 = [\mathbf{z}(t_1), \frac{\partial L}{\partial \mathbf{z}(t_1)}, \mathbf{0}, -\frac{\partial L}{\partial t_1}]$ $\qquad\qquad\qquad\qquad$ ▷ Define initial augmented state

$\quad$ **def** aug_dynamics($[\mathbf{z}(t),\mathbf{a}(t),-,-],t,\theta$): $\qquad\qquad$ ▷ Define dynamics on augmented state

$\qquad$ **return** $[f(\mathbf{z}(t),t,\theta), -\mathbf{a}(t)^\mathsf{T}\frac{\partial f}{\partial \mathbf{z}}, -\mathbf{a}(t)^\mathsf{T}\frac{\partial f}{\partial \theta}, -\mathbf{a}(t)^\mathsf{T}\frac{\partial f}{\partial t}]$ $\qquad$ ▷ Concatenate time-derivatives

$\quad [\mathbf{z}(t_0), \frac{\partial L}{\partial \mathbf{z}(t_0)}, \frac{\partial L}{\partial \theta}, \frac{\partial L}{\partial t_0}] = \text{ODESolve}(s_0, \text{aug\_dynamics}, t_1, t_0, \theta)$ $\qquad$ ▷ Solve reverse-time ODE

**return** $\frac{\partial L}{\partial \mathbf{z}(t_0)}, \frac{\partial L}{\partial \theta}, \frac{\partial L}{\partial t_0}, \frac{\partial L}{\partial t_1}$ $\qquad\qquad\qquad\qquad\qquad$ ▷ Return all gradients

---

## Appendix D  Autograd Implementation

```
import scipy.integrate

import autograd.numpy as np
from autograd.extend import primitive, defvjp_argnums
from autograd import make_vjp
from autograd.misc import flatten
from autograd.builtins import tuple

odeint = primitive(scipy.integrate.odeint)
```

```python
def grad_odeint_all(yt, func, y0, t, func_args, **kwargs):
    # Extended from "Scalable Inference of Ordinary Differential
    # Equation Models of Biochemical Processes", Sec. 2.4.2
    # Fabian Froehlich, Carolin Loos, Jan Hasenauer, 2017
    # https://arxiv.org/pdf/1711.08079.pdf

    T, D = np.shape(yt)
    flat_args, unflatten = flatten(func_args)

    def flat_func(y, t, flat_args):
        return func(y, t, *unflatten(flat_args))

    def unpack(x):
        #      y,        vjp_y,       vjp_t,     vjp_args
        return x[0:D], x[D:2 * D], x[2 * D], x[2 * D + 1:]

    def augmented_dynamics(augmented_state, t, flat_args):
        # Orginal system augmented with vjp_y, vjp_t and vjp_args.
        y, vjp_y, _, _ = unpack(augmented_state)
        vjp_all, dy_dt = make_vjp(flat_func, argnum=(0, 1, 2))(y, t, flat_args)
        vjp_y, vjp_t, vjp_args = vjp_all(-vjp_y)
        return np.hstack((dy_dt, vjp_y, vjp_t, vjp_args))

    def vjp_all(g,**kwargs):

        vjp_y = g[-1, :]
        vjp_t0 = 0
        time_vjp_list = []
        vjp_args = np.zeros(np.size(flat_args))

        for i in range(T - 1, 0, -1):

            # Compute effect of moving current time.
            vjp_cur_t = np.dot(func(yt[i, :], t[i], *func_args), g[i, :])
            time_vjp_list.append(vjp_cur_t)
            vjp_t0 = vjp_t0 - vjp_cur_t

            # Run augmented system backwards to the previous observation.
            aug_y0 = np.hstack((yt[i, :], vjp_y, vjp_t0, vjp_args))
            aug_ans = odeint(augmented_dynamics, aug_y0,
                             np.array([t[i], t[i - 1]]), tuple((flat_args,)), **kwargs)
            _, vjp_y, vjp_t0, vjp_args = unpack(aug_ans[1])

            # Add gradient from current output.
            vjp_y = vjp_y + g[i - 1, :]

        time_vjp_list.append(vjp_t0)
        vjp_times = np.hstack(time_vjp_list)[::-1]

        return None, vjp_y, vjp_times, unflatten(vjp_args)
    return vjp_all

def grad_argnums_wrapper(all_vjp_builder):
    # A generic autograd helper function.  Takes a function that
    # builds vjps for all arguments, and wraps it to return only required vjps.
    def build_selected_vjps(argnums, ans, combined_args, kwargs):
        vjp_func = all_vjp_builder(ans, *combined_args, **kwargs)
        def chosen_vjps(g):
            # Return whichever vjps were asked for.
            all_vjps = vjp_func(g)
            return [all_vjps[argnum] for argnum in argnums]
        return chosen_vjps
```

```
        return  build_selected_vjps
```

```
defvjp_argnums ( odeint ,  grad_argnums_wrapper ( grad_odeint_all ))
```

## Appendix E    Algorithm for training the latent ODE model

To obtain the latent representation $\mathbf{z}_{t_0}$, we traverse the sequence using RNN and obtain parameters of distribution $q(\mathbf{z}_{t_0}|\{\mathbf{x}_{t_i},t_i\}_i,\theta_{enc})$. The algorithm follows a standard VAE algorithm with an RNN variational posterior and an ODESolve model:

1. Run an RNN encoder through the time series and infer the parameters for a posterior over $\mathbf{z}_{t_0}$:

$$q(\mathbf{z}_{t_0}|\{\mathbf{x}_{t_i},t_i\}_i,\phi) = \mathcal{N}(\mathbf{z}_{t_0}|\mu_{\mathbf{z}_{t_0}},\sigma_{\mathbf{z}_0}), \tag{53}$$

   where $\mu_{\mathbf{z}_0},\sigma_{\mathbf{z}_0}$ comes from hidden state of $\text{RNN}(\{\mathbf{x}_{t_i},t_i\}_i,\phi)$

2. Sample $\mathbf{z}_{t_0} \sim q(\mathbf{z}_{t_0}|\{\mathbf{x}_{t_i},t_i\}_i)$

3. Obtain $\mathbf{z}_{t_1},\mathbf{z}_{t_2},\ldots,\mathbf{z}_{t_M}$ by solving ODE $\text{ODESolve}(\mathbf{z}_{t_0},f,\theta_f,t_0,\ldots,t_M)$, where $f$ is the function defining the gradient $d\mathbf{z}/dt$ as a function of $\mathbf{z}$

4. Maximize $\text{ELBO} = \sum_{i=1}^{M} \log p(\mathbf{x}_{t_i}|\mathbf{z}_{t_i},\theta_{\mathbf{x}}) + \log p(\mathbf{z}_{t_0}) - \log q(\mathbf{z}_{t_0}|\{\mathbf{x}_{t_i},t_i\}_i,\phi)$,
   where $p(\mathbf{z}_{t_0}) = \mathcal{N}(0,1)$

## Appendix F    Extra Figures

(a) 30 time points          (b) 50 time points          (c) 100 time points

Figure 10: Spiral reconstructions using a latent ODE with a variable number of noisy observations.

## Footnotes

[1]Note that we've overloaded $t$ to be both a part of the state and the (dummy) independent variable. The distinction is clear given context, so we keep $t$ as the independent variable for consistency with the rest of the text.