[Reviews · NeurIPS 2018]

Reviewer 1



# Response to author feedback My thanks to the authors for their responses to my comments and questions in their feedback and commitment to make several clarifications in response to the suggestions made. After reading their feedback and the other reviewer's comments I maintain my original assessment and feel this is an excellent submission that I'd strongly argue for including in the conference. --- # Summary and relation to previous work This submission makes several related contributions: * it provides a clear and readable introduction to the adjoint sensitivity methods for reverse-mode automatic differentiation (AD) through ODE solvers, * motivated by previous links drawn between Res-Nets and ODE models (Lu, Zhong, Li & Dong, 2017), the direct use of ODE solvers as differentiable (and so learnable) components in machine learning models is proposed, with such 'ODE-Nets' offering advantages of improved memory efficiency during gradient calculation and adpative effective network depth, * continuous normalising flows (CNFs) are proposed as an analogue to previous work on flow models e.g. normalising flows (Rezende & Mohamed, 2015) and NICE (Dinh, Krueger and Bengio, 2015), with CNFs claimed to provided the advantage of allowing more expressive flows while maintaining computational tractability and efficient reversibility of the flow (compared to normalising flows), * a generative latent variable ODE-based time series model (L-ODE) trained with a variational objective is proposed as an alternative to standard recurrent neural network approaches, with more natural handling of irregularly time-sampled sequences. # Strengths and weaknesses Although the adjoint sensitivity method is an existing method, exposing this method to machine learning and computational statistics communities where, as far as I am aware it is not widely known about, is a worthwile contribution of this submission its own right. Given the ever increasing importance of AD in both communities, adding to the range of scientific computing primitives for which frameworks such as autograd can efficiently compute derivatives through will hopefully spur more widespread use of gradient based learning and inference methods with ODE models and hopefully spur other frameworks with AD capability in the community such as Stan, TensorFlow and Pytorch to implement adjoint sensitivity methods. The specific suggested applications of the 'ODE solver modelling primitive' in ODE-Nets, CNFs and L-ODEs are all interesting demonstrations of some of the computational and modelling advantages that come from using a continuous-time ODE mode; formulation, with in particular the memory savings possible by avoiding the need to compute all intermediate states by recomputing trajectories backwards through time being a possible major gain given that device memory is often currently a bottleneck. While 'reversing' the integration to recompute the reverse trajectory is an appealing idea, it would have helped to have more discussion of when this would be expected to breakdown - for example it seems likely that highly chaotic dynamical systems would tend to be problematic as even small errors in the initial backwards steps could soon lead to very large divergences in the reversed trajectories compared to the forward ones. It seems like a useful sanity check in an implementation would be to compare the final state of the reversed trajectory to the initial state of the forward trajectory to check how closely they agree. The submission is generally very well written and presented with a clear expository style, with useful illustrative examples given in the experiments to support the claims made and well thought out figures which help to give visual intuitions about the methods and results. There is a lot of interesting commentary and ideas in the submission with there seeming to be a lot of potential in even side notes like the concurrent mixutre of dynamics idea. While this makes for an interesting and thought-provoking read, the content-heavy nature of the paper and slightly rambling exploration of many ideas are perhaps not ideally suited to such a short conference paper format, with the space constraints meaning sacrifices have been made in terms of the depth of discussion of each idea, somewhat terse description of the methods and results in some of the experiments and in some cases quite cramped figure layouts. It might be better to cull some of the content or move it to an appendix to make the main text more focussed and to allow more detailed discussion of the remaining areas. A more significant weakness perhaps is a lack of empirical demonstrations on larger benchmark problems for either the ODE-Nets or CNFs to see how / if the proposed advantages over Res-Nets and NFs respectively carry over to (slightly) more realistic settings, for example using the CNF in a VAE image model on MNIST / CIFAR-10 as in the final experiments in original NF paper. Although I don't think such experiments are vital given the current numerical experiments do provide some validation of the claims already and more pragmatically given that the submission already is quite content heavy already so space is a constaint, some form of larger scale experiments would make a nice addition to an already strong contribution. # Questions * Is the proposal to backward integrate the original ODE to allow access to the (time-reversed) trajectory when solving the adjoint ODE rather than storing the forward trajectory novel or is this the standard approach in implementations of the method? * Does the ResNet in the experiments in section 7.1 share parameters between the residual blocks? If not a potentially further interesting baseline for the would be to compare to a deeper ResNet with parameter sharing as this would seem to be equivalent to an ODE net with a fixed time-step Euler integrator. * What is the definition used for the numerical error on the vertical axis in Figure 4a and how is the 'truth' evaluated? * How much increase in computation time (if any) is there in computing the gradient of a scalar loss based on the output of `odeint` compared to evaluating the scalar loss itself using your Python `autograd` implementation in Appendix C? It seems possible that the multiple sequential calls to the `odeint` function between pairs of successive time points when calculating the gradient may introduce a lot of overhead compared to a single call to `odeint` when calculating the loss itself even if the overall number of inner integration steps is similar? Though even if this is the case this could presumably be overcome with a more efficient implementation it would be interesting to get a ballpark for how quick the gradient calculation is currently. # Minor comments / suggestions / typos: Theorem 1 and proof in appendix B: while the derivation of this result in the appendix is nice to have as an additional intuition for how the expression arises, it seems this might be more succinctly seen as direct implication of the Fokker-Planck equation for a zero noise process or equivalently from Liouville's equation (see e.g. Stochastic Methods 4th ed., Gardiner, pg. 54). Similarly expressing in terms of the time derivative of the density rather than log density would perhaps make the analogy to the standard change of variables formula more explicit. Equation following line 170: missing right parenthesis in integrand of last integral. Algorithm 1: Looks like a notation change might have lead to some inconsistencies - terms subscripted with non-defined $N$: $s_N$, $t_N$, look like they would be more consistent if instead subscripted with 1. Also on first line vector 0 and scalar 0 are swapped in $s_N$ expression References: some incorrect capitalisation in titles and inconsistent used of initials rather than full first names in some references. Appendices L372: 'differentiate by parts' -> 'integrate by parts' L385: Unclear what is meant by 'the function $j(x,\theta,t)$ is unknown - for the standard case of a loss based on a sum of loss terms each depending on state at a finite set of time points, can we not express $j$ as something like $$ j(x,\theta,t) = \sum_{k=1}^K \delta(t_k - t) \ell_k(x, \theta) $$ which we can take partial derivatives of wrt $x$ and then directly subsitute into equation (19)? L394: 'adjoing' -> 'adjoint'

Reviewer 2



* Summary: This paper proposes a new method to differentiate a loss through ODE solvers. Instead of tracking all operations performed by the solver, it proposes to solve another ODE backwards in time, whose solution brings all derivatives of the loss with respect to the initial values and parameters, thereby achieving memory gains. This leads the authors to propose ODE-Nets, which are networks solving a differential equation parametrized by a neural network. Numerical experiments span classification and generation. For generation, a continuous-time normalizing flow is proposed. Specifically for time-series, a latent generative model allows to take into account irregularly sampled data at continuous times. * Opinion: This is a good paper which provides a good contribution for the community. It is not that original witness previous works on the interpretation of resnets through the lens of dynamical systems, but the reduction in computational time for the backprop still makes it significant. The span of the experiments demonstrate the wide applicability of the introduced method. The ideas are clear overall, except for some parts. There are two weak points: - Experiments are only performed on toy-ish datasets. For the classification example, I would have preferred CIFAR10 to MNIST. - Writing could be improved in some parts; slightly reorganizing the paper would help. Despite these weaknesses, I tend to vote in favor of this paper. * Detailed remarks: - The analysis in Figure 4 is very interesting. What is a possible explanation for the behaviour in Figure 4(d), showing that the number of function evaluations automatically increases with the epochs? Consequently, how is it possible to control the tradeoff between accuracy and computation time if, automatically, the computation time increases along the training? In this direction, I think it would be nice to see how classification accuracy evolves (e.g. on MNIST) with the precision required. - In Figure 6, an interpolation experiment shows that the probability distribution evolves smoothly along time, which is an indirect way to interpret it. Since this is a low (2D) dimensional case, wouldn't it be possible to directly analyze the learnt ODE function, by looking at its fixed points and their stability? - For the continuous-time time-series model, subsec 6.1 clarity should be improved. Regarding the autoencoder formulation, why is an RNN used for the encoder, and not an ODE-like layer? Indeed, the authors argue that RNNs have trouble coping with such time-series, so this might also be the case in the encoding part. - Do the authors plan to share the code of the experiments (not only of the main module)? - I think it would be better if notations in appendix A followed the notations of the main paper. - Section 3 and Section 4 are slightly redundant: maybe putting the first paragraph of sec 4 in sec 3 and putting the remainder of sec 4 before section 3 would help.

Reviewer 3



In this paper, the authors identify ordinary differential equations as the continuous limit of residual networks, and use an adjoint method to derive an augmented ODE which efficiently computes the backward pass. This enables the authors to see ODEs as modeling primitives in deep models, leveraging ODE solvers for efficiently computing both the forward pass and backward pass. This enables the authors to introduce several models: first, they introduce continuous time normalizing flows as a continuous limit of normalizing flows, and show that they are easier to train (no constraint on the transformation, more efficient computation of the probability flow). Second, they investigate time-series latent ODEs, time series parametrized by a deterministic ODE and an initial latent state. The authors combine their methods with VAE-style inference networks to train the resulting model. This is an excellent paper, well written, with exciting methodological work (even if adjoint methods are not new themselves, their application in this context and the leveraging of existing ODE solvers are new ideas in deep learning) and interesting results. I expect the work to have a significant impact. Minor points: - Hopefully the following is somewhat clear: considering Euler's method with a very small step size as a 'layer' of an ODE solver, a single layer changes its input data relatively little, and consequently many layers will be required for a nonlinear transformation. Therefore, having an ODE as a 'flexible' (high modeling power) layer in a network will require many steps of computation, potentially more than using residual networks where each layer accomplishes more 'work' than infinitesimal layers of an ODE. In other words, are ODE as primitive powerful but ultimately more computationally hungry than a network with a similar modeling power (this does not appear to be the case for CNF, but this seems to rely on the concurrent mixture trick which is not available to 'vanilla' NF, so the mixture-CNF does not obviously seem to be the continuous limit of a vanilla NF). - The explanation of the adjoint method in the appendix is somewhat poorly written. It is not generally (or obviously) true that for any Lagrangian multipliers, the lagrangian loss has same gradient as the actual loss J; and the Lagrangian multipliers function cannot be set arbitrarily (as lines 378-379 seem to imply). The discussion on why J might be known but not j is lacking (lines 384-385), and the intervals used in lines 384+ are not defined before being used. - \mathcal{J} used in eq 26 of the appendix does not seem to be defined. - In algorithm box 1, it would be desirable to explicitly provide the expected interface of the function ODEsolve, as well as that of its 'Dynamics' argument (an example is given a few lines above, of course, but the box would be easier to read if everything was spelt out clearly) - Perhaps it would be worth explicitly mentioning that variables z_{t_i} in section 6 are deterministic functions of z__{t_0} (hence having the inference network only infer the 'true' latent z_0) - Could the ideas of section 6 be extended to stochastic differential equations? - The model in 6.1 seems to be missing significant amount of details to connect it to the previous section - which was more general than using poisson process likelihood.